# Targeting Multiple Signaling Pathways in Cancer: The Rutin Therapeutic Approach

**DOI:** 10.3390/cancers12082276

**Published:** 2020-08-14

**Authors:** Zeinab Nouri, Sajad Fakhri, Keyvan Nouri, Carly E. Wallace, Mohammad Hosein Farzaei, Anupam Bishayee

**Affiliations:** 1Student Research Committee, Faculty of Pharmacy, Kermanshah University of Medical Sciences, Kermanshah 6714415153, Iran; zeinab7641@yahoo.com; 2Pharmaceutical Sciences Research Center, Health Institute, Kermanshah University of Medical Sciences, Kermanshah 6734667149, Iran; pharmacy.sajad@yahoo.com; 3Student Research Committee, School of Medicine, Isfahan University of Medical Sciences, Isfahan 8174673461, Iran; keyvan7338@gmail.com; 4Lake Erie College of Osteopathic Medicine, Bradenton, FL 34211, USA; carlywallace96@gmail.com

**Keywords:** rutin, cancer, signaling pathways, therapeutic targets, pharmacology, drug delivery system

## Abstract

Multiple dysregulated signaling pathways are implicated in the pathogenesis of cancer. The conventional therapies used in cancer prevention/treatment suffer from low efficacy, considerable toxicity, and high cost. Hence, the discovery and development of novel multi-targeted agents to attenuate the dysregulated signaling in cancer is of great importance. In recent decades, phytochemicals from dietary and medicinal plants have been successfully introduced as alternative anticancer agents due to their ability to modulate numerous oncogenic and oncosuppressive signaling pathways. Rutin (also known as rutoside, quercetin-3-*O*-rutinoside and sophorin) is an active plant-derived flavonoid that is widely distributed in various vegetables, fruits, and medicinal plants, including asparagus, buckwheat, apricots, apples, cherries, grapes, grapefruit, plums, oranges, and tea. Rutin has been shown to target various inflammatory, apoptotic, autophagic, and angiogenic signaling mediators, including nuclear factor-κB, tumor necrosis factor-α, interleukins, light chain 3/Beclin, B cell lymphoma 2 (Bcl-2), Bcl-2 associated X protein, caspases, and vascular endothelial growth factor. A comprehensive and critical analysis of the anticancer potential of rutin and associated molecular targets amongst various cancer types has not been performed previously. Accordingly, the purpose of this review is to present an up-to-date and critical evaluation of multiple cellular and molecular mechanisms through which the anticancer effects of rutin are known to be exerted. The current challenges and limitations as well as future directions of research are also discussed.

## 1. Introduction

Cancer is a complex and multifaceted disease that is characterized by the unlimited proliferation of abnormal cells with the ability to attack or spread to the whole body [1]. This disease arises as a result of homeostasis imbalance between cell survival and cell death [2]. Multiple signaling pathways involved in the pathogenesis of cancer bolster the need for further research [3,4]. Despite the progress in cancer research, providing more involved pathways and molecular targets is of great importance. Disturbance in the expression of tumor suppressor genes, oncogenes, and apoptotic genes play a key role in the pathophysiological mechanisms of cancer [5,6]. In addition, several inflammatory, oxidative stress, autophagy, and apoptotic dysregulated pathways are involved in the initiation and development of cancer [7,8,9]. A wide variety of intracellular molecules have been identified to provoke the uncontrolled proliferation of cancer cells. For instance, in malignant cells, the upregulation of cyclin-dependent kinase (CDK) and downregulation of tumor suppressor proteins (p53), CDK inhibitors, p21, p27, and p57 have been identified [10]. The inappropriate regulation of signaling proteins, including phosphoinositide 3-kinase (PI3K), protein kinase B (PKB, also known as Akt), mammalian target of rapamycin (mTOR), and mitogen-activated protein kinases (MAPK) as well as the altered expression of various pro-inflammatory transcription factors, including nuclear factor-κB (NF-κB), activating protein-1 (AP-1), hypoxia-inducible factor 1 (HIF-1) and signal transducers and activators of transcription (STAT) families have been reported in tumor cells [11,12,13,14,15]. Chronic inflammation is considered a key driver of both the initiation and progression of tumorigenesis [16]. Therefore, targeting the key aberrant proteins and pathways represents a desirable approach to cancer therapy.

Common forms of cancer treatment include surgery, radiotherapy, stem cell therapy, photodynamic therapy, and chemo/immunotherapies [17,18]. Despite the efficiency of chemotherapy, the demerits associated with classical cytotoxic treatments, including multiple drug resistance (MDR), high financial costs, and severe adverse effects, cause a major hurdle in its clinical application [19]. Thus, there is a dire need to discover new, safe, and more efficacious treatment options to achieve ideal results. Plant-derived natural products have attained great attention in drug discovery programs. Numerous drugs used for cancer therapy, including doxorubicin, vinblastine, paclitaxel, and camptothecin, have been obtained from natural sources [20]. The use of chemo-herbal combination therapy has been found to increase the anticancer effects of chemotherapeutic agents and to ameliorate drug resistance and chemotherapy-related adverse effects [21,22]. Natural secondary metabolites have shown pleiotropic effects and target various cancer hallmarks, including inflammation, cancer cell proliferation, migration, invasion, angiogenesis, and metastasis [23].

As natural compounds are potential multi-targeted agents in combating cancer, they are of great interest to prevent associated side effects in treating cancer [24]. Oxidative stress and inflammation associated with synthetic anticancer agents are implicated in high levels of toxicity, host tissue damage, and even manifestation of secondary tumors [25]. Growing evidence demonstrates that cytostatic effects of natural products are derived from their potential in modulating oxidative stress, inflammation, autophagy and apoptosis, thereby leading to the prevention/reduction of their associated toxicity [26,27]. Indeed, free radical generation and pro-oxidant properties of natural agents seem to underlie their direct toxicity towards tumor cells. At the same time, antioxidant properties of naturally occurring agents contribute to their cancer preventive ability and lower toxicity compared to synthesized anticancer drugs [25].

Rutin, also known as rutoside, quercetin-3-O-rutinoside and sophorin, is a glycoside consisting of the flavonol quercetin and the disaccharide rutinose. Rutin has also been called vitamin P, as it is widely distributed in various plants, from vegetables and fruits to medicinal plants, including asparagus, buckwheat, apricots, apples, cherries, grapes, grapefruit, plums, oranges, and tea. Rutin has shown ubiquitous pharmacological properties, including antioxidant, anti-inflammatory, antiangiogenic, pro-apoptotic, and antiproliferative activities, all of which may participate in the prevention and treatment of cancer [28,29,30,31,32,33,34]. Figure 1 displays the chemical structure of rutin and its reversible deglycosylation to produce quercetin. Amongst previous reviews, Prasad et al. [35] and Ganeshpurkar et al. [36] described the pharmacological activities of rutin in combating several diseases, with very limited information related to cancer studies. In another study, Perk et al. [37] reviewed the anticancer effect of rutin without a specific focus on cancer types. A comprehensive and critical analysis of the anticancer effects of rutin and associated molecular targets amongst various cancer types has not been performed before. Therefore, the purpose of this review is to present an up-to-date and critical evaluation of multiple cellular and molecular mechanisms through which the anticancer effects of rutin are known to be exerted. The current challenges and limitations, as well as future directions of research, are also discussed.

## 2. Role of Inflammation, Oxidative Stress, Apoptosis, and Autophagy in Cancer Progression

Prevailing studies are revealing the critical roles of inflammation, oxidative stress, apoptosis, and autophagy in cancer progression. A growing body of evidence has shown that inflammatory responses are key components of tumorigenesis and cancer promotion [38]. Several inflammatory mediators, including cytokines, such as interleukin-1β (IL-1β), IL-6, tumor necrosis factor-α (TNF-α), chemokines, growth factors, and reactive oxygen species (ROS) contribute to the proliferation, metastasis, angiogenesis, and chemoresistance of cancer cells by activation of transcription factors like MAPKs, NF-κB, and mTOR [39]. Aberrant regulation of the molecular pathways involved in inflammation displays a close association with cancer [40]. Targeting impaired inflammatory molecules represents an attractive approach for cancer therapy [41]. Upon exposure to stressful stimuli, the upregulation of the aforementioned intracellular signaling pathways triggers the synthesis and release of inflammatory cytokines, oxidative stress, and carcinogenesis [42]. Constitutive activation of MAPKs and NF-κB signaling pathways have been reported in several types of cancers [41]. MAPKs include a family of protein serine/threonine kinases, which are classified into three main subfamilies, including c-Jun NH2-terminal kinase (JNK), extracellular signal-regulated kinase (ERK) and p38 [43]. The MAPKs signaling cascade plays a critical role in inflammation-associated cancers, participating in cell proliferation, differentiation, and apoptosis [13]. As a parallel pathway to MAPKs, and an upstream signaling pathway of NF-κB, the PI3K/Akt/mTOR signaling pathway is also linked with the regulation of inflammation and cancer cells survival [44]. Accumulating data suggest that there is crosstalk between mTOR activation and inflammatory response, which contributes to the coupling of cell survival and proliferation in response to environmental stimuli [45]. Accordingly, mTOR interacts with the upstream molecule mesenchymal–epithelial transition factor (c-met) to regulate tumor progression. Upregulation of c-met and its ligand, hepatocyte growth factor (HGF), provokes PI3K/Akt/mTOR, Ras/Raf/mitogen-activated protein kinase kinase (MEK)/ERK/MAPK, paxillin/Ras-related C3 botulinum toxin substrate 1 (Rac-1), and STATs signaling cascades, thereby causing inflammation, proliferation, migration, angiogenesis, and metastasis [46,47]. Furthermore, malignant cells activate PI3K/Akt/mTOR, Ras/Raf/MEK/ERK/MAPK, and AP-1/vascular endothelial growth factor (VEGF) pathways via growth factor binding to their putative receptors such as insulin-like growth factor receptor (IGFR), platelet-derived growth factor receptor (PDGFR), and epidermal growth factor receptor (EGFR) [48].

Compelling studies have also demonstrated that ROS play a fundamental role in crosstalk between autophagy and apoptosis [49,50]. Oxidative stress, resulting from an imbalance between ROS production and elimination by enzymatic/non-enzymatic antioxidants including superoxide dismutase (SOD), catalase (CAT), glutathione peroxidase (GPx), and glutathione (GSH), promotes tumor cell proliferation, angiogenesis, and metastasis. Surprisingly, new evidence indicates that ROS are not only able to induce tumorigenesis but also possess tumor-suppressive properties [51]. Considering the dual role of ROS in cancer, nuclear factor erythroid 2–related factor 2 (Nrf2)-mediated antioxidant response may act as an anti- or as a pro-tumorigenesis [52]. It has been reported that ROS amplify transcription factor AP-1 which, in turn, augments VEGF expression, a trigger of the angiogenesis cascade.

Apoptosis or type I programmed cell death plays a central role in the pathogenesis of cancer. As a hallmark of cancer, apoptosis resistance leads to uncontrolled proliferation, cancer cells survival under hypoxic conditions, and resistance to chemotherapeutic drugs [53]. The underlying mechanisms by which cancer cells evade apoptosis encompasses the downregulation of pro-apoptotic proteins, upregulation of anti-apoptotic proteins, and the dysregulation of death receptors and p53-related signaling pathways [54]. Apoptosis can occur through two well-known apoptotic pathways, including the mitochondrial pathway (intrinsic) and the death receptor pathway (extrinsic) [55]. The intrinsic and extrinsic pathways are associated with caspase-9 and caspase-8, respectively. The extrinsic pathway is initiated through the occupation of cell surface death receptors of the TNF receptor family and the intrinsic pathway is triggered by cellular stresses [56,57]. Tumor suppressor p53 can regulate the intrinsic pathway of apoptosis through controlling the B cell lymphoma 2 (Bcl-2) proteins family [58]. Induction of pro-apoptotic BH3-only proteins suppresses the pro-survival Bcl-2 proteins, thereby leading to the upregulation of pro-apoptotic proteins BH3-interacting domain death agonist (Bid), Bcl-2 antagonist killer (BAK), and Bcl-2 associated X protein (Bax) which, in turn, cause the outer mitochondrial membrane to become permeable and the release of cytochrome c from the mitochondria [59]. Cytochrome c then activates cysteine protease enzymes called caspases that are responsible for cleaving vital cellular proteins [60].

On the other hand, p53 possesses the ability to control the expression of components of the death receptors pathway. P53 activates the death receptors tumor necrosis factor receptor 1 (TNFR1) and fatty acid synthase (FAS) to sensitize cells to death ligands TNF-α, Fas ligand (FasL), and TNF-related apoptosis-inducing ligand (TRAIL), thus facilitating apoptosis [61]. Interestingly, MAPK kinase 4 (MKK4), selective for JNK activation, couples oncogenic stimuli to p53 activation which, in turn, leads to p21-mediated cell-cycle arrest and/or Bax-mediated apoptosis [62]. TNF-α not only participates in fostering tumor growth through chronic inflammation but also amplifies apoptosis through activating the extrinsic pathway. NF-κB also serves as a key factor in inducing apoptosis mediated by TRAIL or TNF-α [63]. Therefore, impaired activation of NF-κB expedites resistance to apoptosis. TNF-α also promotes poly (ADP ribose) polymerase (PARP) activation [64], an important enzyme in DNA repair and programmed cell death. PARP inhibitors expedite ROS production, DNA damage, and programmed cell death [59]. Phosphatases and tensin homolog (PTEN) is an important tumor suppressor gene that negatively regulates the PI3K/Akt/mTOR anti-apoptotic pathway. Impairment of the PTEN/ PI3K/Akt/mTOR pathway represses apoptosis and promotes tumorigenesis [65].

Autophagy (programmed cell death type II) is an intracellular regulated process that plays a vital role in the maintenance of cellular homeostasis by eliminating malformed and unwanted proteins [66]. Aberrant regulation of autophagy contributes toward tumorigenesis. Autophagy acts as a double edged sword, containing both tumor suppression and tumor promotion characteristics [67]. This dual role of autophagy poses a great challenge in the development of efficient anticancer drugs. As a homeostasis control process, autophagy displays cytoprotective properties through the degradation of misfolded proteins and the clearing of ROS. As a tumor promoter, the stress-hindering activities of autophagy protect malignant cells from necrosis caused by metabolic stress. Autophagy also supplies the elevated energy demands of tumor cells, which is necessary for tumor cell survival and proliferation [68]. Mechanistically, JNK, p38MAPK, and ERK signaling pathways positively regulate autophagy in malignant cells. Upregulation of these pathways putatively activates autophagy associated proteins like autophagy-related protein (Atg), Beclin1, and light chain 3 (LC3B) [69,70,71]. MTOR is a well-known inhibitor of autophagy. As upstream regulators of mTOR, Akt and Forkhead box O3 (FoxO3) play key roles in the positive regulation of mTOR and inhibition of autophagy [72].

In summary, there are numerous altered signaling pathways identified across several cancer types. Targeting cross-linked intracellular signaling pathways that are associated with dysregulated proliferation and cell survival by utilizing multi-targeted agents is an attractive strategy to combat cancer.

## 3. Rutin: Sources and Pharmacological Effects

Rutin (3, 3′, 4′, 5, 7-pentahydroxyflavone-3-rhamnoglucoside) is a flavonol glycoside found in a wide variety of vegetables, fruits, and beverages, including passionflower, grapes, green asparagus, apples, tea, and wine. A large number of medicinal plants also contain rutin, such as Buckwheat (*Fagopyrum esculentum* Moench), *Ruta graveolens* L., *Sophora japonica* L., *Maranta leuconeura* E. Morren., and *Eucalyptus* spp. [73,74,75], with the former being the most significant source of natural rutin [76]. Rutin has been also isolated from several herbal families, including Polygonaceae, Rutaceae, Fabaceae, Marantaceae, and Myrtaceae [75,77]. It has been reported that the concentration of rutin varies within the different parts of plants, becoming elevated after UV-B exposure to protect against radiation [78,79]. Rutin is also called vitamin P or rutoside and contains extensive pharmacological properties, including neuroprotective [30], hepatoprotective [80,81], cardioprotective [82], and anticarcinogenic activities [37]. Rutin has also been demonstrated to hamper inflammation, oxidative insults, and platelet aggregation [83]. The insulin-sensitizing and lipid-lowering properties of rutin support the beneficial effects of this agent in diabetes mellitus, hyperlipidemia, and cardiovascular disease. The underlying mechanisms by which rutin counteracts diabetes and its complications include the suppression of gluconeogenesis, increased glucose uptake, and the abrogation of intestinal glucose absorption [84]. Rutin also reverses endothelial dysfunction through enhancing nitric oxide production and repressing ROS responsive nucleotide-binding domain-like receptor 3 (NLRP3) [85,86], thereby decreasing the risk of cardiovascular disease. Rutin has been also reported to combat neurodegenerative diseases by abrogating neuroinflammation, abnormal protein accumulation, and apoptosis, as well as regulating microglia and astrocyte activation [87,88,89]. It has been documented that rutin possesses promising nephroprotective effects against nephrotoxins, such as cisplatin, vancomycin, and mercuric chloride, via mitigating inflammation, oxidative damage, apoptosis, and enhancing aquaporin 1 level [90,91,92]. From another mechanistic perspective, rutin also targets several inflammatory mediators such as NF-κB and TNF-α, thereby counteracting inflammation-driven disease. The hepatoprotective properties of rutin in animal models of non-alcoholic fatty liver disease include its ability to mitigate autophagy corroborated by abrogating key autophagy biomarkers and modulating the expression of lipolytic and lipogenic genes [93]. In various preclinical models, rutin has been also shown to elevate Nrf2 accompanied by an increase in enzymatic/non-enzymatic antioxidant activities, including SOD, CAT, and GPx, thereby alleviating the aforementioned diseases.

Rutin can, overall, be regarded as a promising multi-targeted nutraceutical agent that elicits several health benefits.

## 4. Methodology for Literature Search on Rutin and Cancer

The present systematic review was performed according to the Preferred Reporting Items for Systematic Reviews and Meta-Analysis (PRISMA) criteria. PRISMA statement is useful for improving the reporting of systematic reviews and meta-analyses [94]. A systematic literature search was performed using the scholarly electronic database, including PubMed, Science Direct, and Scopus. The last search was made in June 2020. The systematic search in databases was conducted using the following keywords: “Rutin” and (“cancer” OR “neoplasm” OR “malignancy” OR “carcinoma” OR “melanoma” OR “leukemia” OR “tumor”) [full text]. It should be mentioned that in the Scopus database, the aforementioned keywords were found in [title/abstract/keywords]. Out of the initial 2113 articles that were obtained by electronic search, 737 were excluded due to duplicated results, 71 were excluded because they were reviews, and 919 were irrelevant based on title and/or abstract information. Additionally, 28 were omitted since they were not in English. Among 358 retrieved articles, 125 were excluded as they evaluated other pharmacological effects of rutin rather than anticancer effects and 161 were ruled out since they focused on other compounds, not rutin. Finally, 72 reports were included in this review, as shown in a summary of results in Figure 2.

## 5. Anticancer Activities of Rutin

Rutin has been found to counteract several types of cancer through various mechanisms, e.g., inhibition of malignant cell growth, induction of cell cycle arrest and apoptosis, and modulation of angiogenesis, inflammation, and oxidative stress, all of which are mediated by regulating multiple cellular signaling pathways. The pharmacological activities and basic antitumor mechanisms of rutin in several cancer types are discussed below.

### 5.1. Rutin and Breast Cancer

Breast cancer is a multifaceted and heterogeneous disease [95]. Based on the presence or absence of three molecular biomarkers, estrogen receptor-α (ER-α), progesterone receptor (PR), and human epidermal growth factor-2 (HER2), breast cancer is classified into five distinct molecular subtypes: (a) luminal A (positive for ER-α and/or PR while negative for HER-2); (b) luminal B (positive for ER-α and/or PR as well as HER2); (c) HER-2 overexpressing; (d) triple-negative; and (e) normal breast-like tumors [96,97]. Triple-negative breast cancer (TNBC) is a heterogenetic and aggressive subtype of breast cancer that is negative for the expression of ER-α, PR, and HER2 [98]. TNBC represents poor prognosis and outcome due to the lack of ideal target options [99]. Therefore, there exists a dire need to discover new targeted therapies for counteracting TNBC. Overactivation of c-met and its ligand, HGF, plays a key role in the initiation and/or progression of TNBC [100]. It has been reported that c-met/HGF is involved in inducing several downstream effectors of different signaling pathways such as Ras/Raf/MEK/ERK/MAPK, PI3K/Akt/mTOR, and Rac-1 [46,47]. Targeting c-met/HGF signaling with novel inhibitory agents is an innovative strategy to combat TNBC. Rutin exhibits anticancer effects on TNBC cell lines through abrogating c-met/HGF axis and its downstream cascades, including paxillin, Rac-1, mTOR, and Akt [101] (Table 1). Additionally, rutin was capable of decreasing the average tumor volume of the TNBC in nude mice [101]. Rutin is therefore a promising c-met inhibitor that may serve as a suitable option to hamper c-met-dependent malignancies.

P53 is a well-known tumor suppressor gene that participates in the induction of cell cycle arrest and apoptosis [158]. Upregulation of p53 promotes p21 activation and subsequently leads to the abrogation of a myriad of cell cycle proteins, including CDK6, CDK2, CDK4, and cyclin B1 [159,160]. Rutin promotes cell cycle arrest at the G2/M phase through interfering with p53- and p21-dependent pathways in ER-α positive-breast cancer MCF-7 cells (luminal A subtype). Additionally, rutin markedly induces apoptosis through enhancing p53 and PTEN. Rutin synergistically increases the antiproliferative effect of tamoxifen on ER-α positive-breast cancer MCF-7 cells [102]. Therefore, rutin may be considered a promising adjuvant agent to increase tamoxifen efficacy in ER-α positive-breast cancer.

From another mechanistic point of view, hypercholesterolemia plays a key role in the progression of breast cancer [161]. Elevated cholesterol levels are associated with uncontrolled cell growth and a worse breast cancer prognosis [162]. The upregulation of FAS participates in tumorigenesis by hampering apoptosis [163]. Rutin abates FAS, elevates antioxidants, and causes cytotoxicity in MCF-7 cells by inducing caspase-dependent apoptosis [103]. Additionally, rutin illustrates anticancer effects against Ehrlich ascites carcinoma, an animal model of breast cancer, as observed by mitigating carcinoembryonic antigen, tumor volume, and cholesterol levels [103].

Prolonged chemotherapy often leads to MDR, which is implicated in the failure of conventional chemotherapeutic agents [164]. MDR occurs due to the upregulation of several drug efflux transporters and the failure of apoptotic pathways. Targeting adenosine triphosphate-binding cassette (ABC) transporters including P-glycoprotein (P-gp/ABCB1), breast cancer resistance protein (BCRP/ABCG2), and multidrug resistance-associated protein-1 (MRP1/ABCC1) by natural products has been a critical approach to reverse MDR and restore chemosensitization [165,166]. Chemoresistance to anticancer therapy is the main cause of tumor recurrence [167]. Therefore, abrogation of chemoresistance can mitigate the relapsed tumor. As a chemosensitizing agent, rutin can be considered as a promising nutraceutical agent to alleviate relapsed tumors. It has been found that the formulations containing rutin and other compounds (arctigenin, arctiin, berberine, berbamine, sanguinarine, and chelerythrine) can successfully inhibit the tumor resistance to chemotherapy, thereby preventing tumor recurrence [168]. Hydrolyzed rutin, a compound modified via rutin deglycosylation, displayed antiproliferative effects and diminished anaplasia in a mouse model with recurrent glioblastoma [169]. Interestingly, rutin amplifies chemosensitivity to cyclophosphamide and methotrexate while reversing MDR by suppressing P-gp and BCRP pumps in MB-MDA-231 and MCF-7 cell lines using well-characterized models of TNBC and HER2-negative breast cancer, respectively. From a different anticancer mechanistic perspective, rutin arrests the cell cycle at G2/M and G0/G1 phases, thereby inducing cell apoptosis [104]. Rutin diminished the resistance to doxorubicin in MCF-7/ADR cells [105]. In addition, rutin showed the potential to suppress angiogenesis, VEGF synthesis and expression in MDA-MB-231 breast cancer cells [106]. Interestingly, this phytochemical depicted antitumor effect via cell cycle arrest at S phase and ROS-mediated apoptosis in MCF-7 cells [107]. Rutin-vanadium complex successfully provoked apoptosis through interfering with p53, Bax, Bcl-2 and abated VEGF expression in both MCF-7 and MDA-MB-231 cells [108]. Further research is needed to confirm the potential of rutin as an adjuvant or synergistic agent in breast cancer therapy.

Controlled release systems are a promising strategy to decrease the fluctuation of drug concentration, enhance treatment efficacy, and diminish side effects [170]. Fabrication of hydrogels of both natural and synthetic polymers offers various advantages, as they supply controlled release and targeting, protect incorporated drugs from degradation and metabolism, and exhibit good biocompatibility and biodegradable properties [171,172]. The pH-responsive hydrogels incorporated with rutin and 5-fluorouracil were successfully formulated using natural water-insoluble polymer (Zein) with the synthetic monomer (acrylic acid). The anticancer effect was evaluated against MDA-MB-231 and MCF-7 breast cancer cell lines. Rutin and 5-fluorouracil loaded pH-sensitive Zein-co-acrylic acid hydrogels demonstrate a controlled release manner and augment anticancer effects by inducing apoptosis and ROS generation [109]. Based on these results, pH-sensitive hydrogels may be a suitable formulation for oral delivery of anticancer drugs with the intent of attaining the tumor site-responsive controlled release and thereby decreasing undesirable toxic effects in normal tissues.

### 5.2. Rutin and Lung Cancer

Lung cancer is the most frequent leading cause of cancer-related death worldwide [173]. Distant metastasis, resistance to the chemotherapeutic regimens, and the cytotoxicity of the drugs are common causes of death amongst lung cancer patients [174]. Therefore, there exists an urgent need to discover non-toxic alternative treatments for chemotherapy responsive lung cancer. In this regard, Wu et al. [110] revealed that rutin exhibits cytotoxicity against A549 human lung cancer cells through modulating TNF-α and glycogen synthase kinase-3β (GSK-3β) expression. GSK-3β participates in numerous cellular processes including proliferation, the cell cycle, and apoptosis [175]. Fibronectin and collagen type I and IV play an important role in the formation of the extracellular matrix, which controls adhesion and migration of cancerous cells [176]. Rutin hampers the adhesion of A549 cells to Fibronectin and collagen type I and IV, thereby inhibiting the migration of lung cancer cells. Additionally, rutin enhances ROS generation and alleviates superoxide production in A549 cells [29]. Rutin hindered the increased effect of β-carotene on single-strand DNA break induced by 4-(methylnitrosamino)-1-(3-pyridyl)-1-butanone in A549 cells. This effect can be ascribed to its antioxidant properties since it abolished ROS level [111]. Rutin repressed lung metastasis induced by B16FlO melanoma cells in mice as observed by decreasing the lung tumor nodules and enhancing the life span of mice [112]. Similarly, in another in vivo study, rutin diminished the number of metastatic nodules, growth, and invasion index, thereby ameliorated lung metastasis induced by B16FlO melanoma cells in mice [113]. From another mechanistic point of view, rutin induces autophagy in A549 cells corroborated by elevating Beclin1, Atg5/12, and LC3-II expression. Additionally, rutin mitigates the expression of NF-κB and TNF-α, acting as a modulator of tumorigenesis [114]. Several transcription factors, namely NF-κB and STAT, have been identified as direct targets of p38 [177]. P38 is then phosphorylated and activated by MKK3 and MKK6 [178], inducing inflammation by producing various pro-inflammatory mediators, such as IL-1β, TNF-α, cyclooxygenase (COX)-2, and inducible nitric oxide synthase (iNOS) [177]. The majority of existing data suggests opposing activity of the p38 signaling pathway with respect to apoptosis and cell cycle modulation [179]. In an in vitro study, rutin prevented the development of lung cancer by diminishing NF-κB and p38 expression, arresting the cell cycle [115]. Additional experiments on lung cancer cell lines and in vivo tumor models should be conducted to further evaluate the beneficial effects of rutin against lung cancer.

### 5.3. Rutin and Colon Cancer

Colorectal cancer results from various risk factors, such as inflammatory bowel disease, obesity, and smoking [180]. Dietary rutin considerably abolishes the viability of human colon adenocarcinoma HT 29 cells in a concentration-dependent manner. Rutin-mediated inhibition of HT 29 cells is achieved by augmentation of cleaved caspase-3, caspase-8, caspase-9, and PARP. PARP is an important enzyme in the detection of DNA damage and programmed cell death [181]. Additionally, rutin upregulates Bax and downregulates Bcl-2. These findings illustrate that rutin induces apoptosis in HT 29 colon cancer cells through concomitant activation of the death receptors and mitochondrial pathways [116].

VEGF is considered a key factor in angiogenesis and tumor growth promotion. Therapeutic intervention involving the inhibition of VEGF has become an innovative strategy for abrogating tumor metastasis [182]. Rutin exerts cytotoxic effects against SW480 colon cancer cells in vitro, markedly suppressing tumor growth and diminishing the expression of VEGF in vivo [117]. In another study, combined treatment of rutin and irradiation sensitized the HT-29 cells to irradiation. Further, concurrent rutin treatment enhanced apoptotic cells, DNA damage, and lipid peroxidative markers. The antioxidant performance elicited by concurrent rutin treatment was reduced by inhibiting antioxidant enzymes (SOD and CAT) and decreasing the mitochondrial membrane potential as cell survival and apoptosis factor [118]. Therefore, rutin is a suitable candidate to increase the radiotherapy response to colon cancer. In an in vitro study, rutin depicted cytotoxic activity on HT-29 cells via increasing ROS generation, ameliorating superoxide production, impairing cell adhesion, and mitigating migration [29].

There is increasing evidence that inflammation is implicated in the proliferation, survival, invasion, angiogenesis, and metastasis of tumor cells [183,184]. Targeting the inflammatory signaling pathway by rutin provides an attractive strategy for cancer prevention and treatment. Rutin effectively ameliorates the expression of biomarkers of the NF-κB inflammatory pathway, including NF-κB, IκB kinase (IKK)-α, and IKK-β in HT-29 colon cancer cells [119]. This indicates that rutin may play a critical role in the prevention of inflammation-mediated cancers. MAPKs are involved in modulating numerous cellular activities related to cancer progression, including inflammatory cascades, proliferation, differentiation, and apoptosis [13], supporting the use of potent MAPKs inhibitor agents. Rutin hinders tumor growth in vitro through interfering with p38MAPK and MAPK activated protein kinase 2 (MK-2). Moreover, rutin ameliorates apoptosis by targeting apoptosis-related proteins, including caspase-3, caspase-8, caspase-9, Bax, Bcl-2, and p53 [119].

Dysregulated metabolism contributes to tumor initiation and progression [26,185]; therefore, their regulation in cancer is of great importance. Rutin ameliorates the metabolism of colon cancer SW480 cells, increases apoptosis, and arrests the cell cycle at the sub-G1 phase. Analysis of microRNAs, long noncoding RNAs, messenger RNAs, and transcription factors revealed that these promising effects were associated with the modulation of dysregulated intracellular signaling pathways involved in glucose, lipid, and protein metabolism, extrinsic and intrinsic apoptosis, reticulum stress responses, and cell cycle stages [120]. Future studies should investigate the proposed panel in other cancer models. Rutin encapsulated in low methoxyl pectin beads abolished cell viability of HT-29 colon cancer cells [121]. It presented antitumor effects via cell cycle arrest at S phase and ROS-mediated apoptosis in LoVo colon cancer cells [107]. Rutin also protected colon cancer Caco2 cells against hydrogen peroxide-induced DNA damage; however, it did not enhance the DNA repair process [122,123]. Deschner et al. [124] indicated the potential of rutin in repressing azoxymethanol (AOM)-induced colonic neoplasia as seen by decreasing focal areas of dysplasia and abrogating hyperproliferation of colonic epithelial cells. In another in vivo study, rutin hindered aberrant crypt foci and induced apoptosis in AOM-induced rat colon cancer [125]; however, Dihal et al. [126] showed that rutin in contrast to its aglycone, quercetin, exerted no protective effect against AOM-induced colorectal carcinogenesis in rats. In this line, rutin did not hamper methylcholantrene (MCH)-mediated CYP1A1 activation, as an enzyme metabolizing precarcinogenic agents, participates in carcinogenesis of intestinal cells (HCT-8) [127]. In another study, rutin could not hinder the development of AOM-induced rat colon cancer and augmented tissue inhibitor of metalloproteinase 1 (TIMP-1) expression, a biomarker of colorectal cancer progression [128]. Overall, further biological and biochemical effects of rutin in colon cancer are needed in-depth clarification in future studies.

### 5.4. Rutin and Brain Cancer

Due to low targeting and negligible permeability of anticancer agents through the blood–brain barrier, brain cancer is an aggressive and devastating neoplasm that is difficult to treat [186]. As a part of the MAPK family, ERK is aberrantly upregulated in cancers expediting the survival, proliferation, and migration of cancer cells [187]. Additionally, ERK participates in crosstalk between programmed cell death and autophagy [188]. ERK plays a key role in TNF-induced autophagy, inhibition of which enhances cellular sensitivity to TNF-induced apoptosis [189]. Discovering and developing new agents to hinder ERK activity is a promising anticancer strategy. Rutin displays pro-apoptotic and antiproliferative effects on human glioblastoma cell lines (GL-15) by diminishing the level of ERK1/2 phosphorylation. Additionally, rutin stops the cell cycle at the G2 stage and stimulates differentiation of GL-15 cells towards an astroglial phenotype, characterized by the upregulation of glial fibrillary acidic protein (GFAP), an astrocyte neurobiomarker [129]. Rutin inhibited the invasion and angiogenesis of GL-15 cells corroborated by mitigating the VEGF and transforming growth factor (TGF)-β1 [130]. In another study, it also exerted an antiproliferative effect on GL-15 cells accompanied by an anti-invasive activity with regard to the potential of this nutraceutical agent in decreasing metalloproteinase (MMP-2) expression, as well as enhancing the expression of extracellular matrix proteins including fibronectin and laminin [131].

MYCN oncogene is a characteristic feature of an advanced and aggressive neuroblastoma stage, representing a poor prognosis [190]. MYCN is a desired target for counteracting neuroblastoma. Rutin obviously abrogates MYCN expression and suppresses the migration and invasion of human neuroblastoma cells, LAN-5. Rutin promotes apoptosis corroborated by a decrease of Bcl-2 expression and Bcl-2/Bax ratio. Additionally, rutin blocks cell cycle progression at the G2/M stage and ameliorates inflammation through the attenuation of TNF-α secretion [132]. Therefore, rutin may be considered a suitable candidate for the treatment of MYCN-dependent tumors.

Rutin exhibits an apoptotic effect on human glioma CHME cells through inducing ROS generation and abating mitochondrial membrane potential. A promising apoptotic effect of rutin was further corroborated by the upregulation of p53, caspase-3, caspase-9, cytochrome c, and Bax as well as the downregulation of Bcl-2 [133]. In addition to the critical role of apoptosis in preventing cancer, autophagy plays an important role in the maintenance of cellular homeostasis and metabolism management [191]. Autophagy can represent both oncogenic and cancer suppressive features, thereby acting as a double-edged sword in cancer cells [192]. Considering the dual function of autophagy in tumorigenesis, both suppression and promotion of this pathway have attracted attention as a promising cancer treatment. The JNK pathway exhibits a multifaceted role in regulating autophagy, apoptosis, and DNA damage [193]. Rutin interestingly mitigates JNK activity both in vitro and in vivo, thereby amplifying the cytotoxic effect of temozolomide through blocking JNK-mediated autophagy. Rutin reinforces the apoptosis effect of temozolomide corroborated by the overexpression of cleaved caspase-3 [134].

### 5.5. Rutin and Leukemia/Multiple Myeloma/Lymphoma

Acute myeloid leukemia is a heterogeneous and aggressive malignancy characterized by the accumulation of immature myeloid hematopoietic cells [194]. The pivotal role of GSK-3β in preserving quiescent hematopoietic stem cells makes it a promising therapeutic target in acute human leukemia [195]. Rutin triggers apoptosis of leukemic cells and promotes cell quiescence through activating Akt and inhibiting GSK-3β. Rutin suppresses the survival of adherent leukemic cells, thus it may be considered as a promising therapeutic agent to combat cell adhesion-mediated drug resistance in acute myeloid leukemia [135]. Rutin decreases tumor weight and volume in human leukemia HL-60 cells in a murine xenograft animal model [136]. In another in vivo study, rutin reduces liver/spleen weight, abolishes proliferation, and augments the activity of macrophage phagocytosis, thereby inducing an immune response in WEHI-3-induced leukemia model in BALB/c mice [137]. Interestingly, rutin exhibits anticancer effects on the leukemia THP-1 cells by promoting autophagy and diminishing inflammation corroborated by decreasing NF-κB and TNF-α [114]. Belonging to the c-Jun subfamily, AP-1 is a well-known transcription factor that plays a critical role in the positive regulation of VEGF [196]. AP-1 activity can be regulated by transcription factors, such as ERK, p38, and JNK [197]. ROS promotes VEGF as a trigger of the angiogenesis cascade [198]. On the other hand, activation of insulin-like growth factor 1 receptor (IGF-1R)/insulin receptor substrate-1 (IRS-1) signaling pathway amplifies the activity of AP-1, which, in turn, stimulates VEGF expression [199]. Targeting the VEGF signaling pathway by naturally occurring compounds appears to be a promising antiangiogenic approach to combat tumor growth. Rutin and vitamin E synergistically suppress VEGF in HL-60 cells. This beneficial effect was mainly attributed to the downregulation of AP-1 and IGF-1R/IRS-1. Antioxidant activity from a combined treatment of rutin and vitamin E (confirmed by decreasing ROS generation) plays a partial role in the decrease of VEGF secretion [138]. Rutin-zinc complex contains antioxidant and cytotoxicity activity against leukemia (KG1) and multiple myeloma (RPMI8226) cell lines [139]. ROS scavenging properties of rutin caused a protective effect of this compound against hydrogen peroxide-induced single-strand DNA break in human myelogenous leukemia cells (K562) [140]. Rutin exhibited anticancer effect through reinforcing susceptibility of K562 cells to natural killer cell-mediated apoptosis [141]. However, Shen et al. [142] revealed that rutin exhibited no apoptosis effect in human promyeloleukemic HL-60 cells compared to its aglycone, quercetin. On the other hand, rutin combined with cytarabine decreased the antiproliferative effect of cytarabine in L1210 leukemia cells [143]. In another study, rutin caused a cytotoxicity in ARH–77 multiple myeloma cell line and mitigated mitochondrial and lysosomal activity [144]. Rutin promotes apoptosis and abrogates GSH levels in Dalton’s lymphoma cells. According to a molecular docking study, rutin acts as a potential suppressor of anti-apoptotic proteins, namely Bcl-xL and cellular FLICE-inhibitory protein (c-FLIP), and antioxidant enzymes, such as GST and glutathione reductase [145]. Further studies should be performed to verify the in silico results.

### 5.6. Rutin and Liver Cancer

Chronic liver diseases, including persistent viral hepatitis and alcoholic and nonalcoholic fatty liver disease are common causes of liver cancer [200]. Approximately 90% of liver cancers are recognized as hepatocellular carcinomas (HCCs) and 10% are cholangiocarcinomas (CCAs) [200]. Due to the asymptomatic feature of this disease, a diagnosis is made at an advanced stage and therefore therapeutic approaches remain ineffective [201]. A deeper exploration of the biology of HCC and CCA, regarding the development of potential therapies, is desperately needed. Rutin induces DNA damage, suppresses uncontrolled proliferation, and decreases cell viability in HTC hepatic cells. Additionally, rutin exhibits a protective effect against the procarcinogenic agent, benzopyrene, through decreasing DNA damage [146]. In another study, rutin dramatically promoted early/late-stage apoptosis and mitigated proliferation, invasion, and colony formation of HEPG2 cells [147]. An imbalance between phase 1 and phase 2 metabolism implicates toxicity through oxidative insults. Agents that hinder phase 1 metabolism, such as cytochrome P450-produced reactive intermediates, or that augment phase 2 metabolism, such as antioxidant enzymes, are considered potential protective agents against chemical carcinogenesis [202]. Treatment by rutin abrogates cytochrome P450-dependent CYP3A4 and CYP1A1 enzymes in addition to enhancing the antioxidant enzymes NADPH Quinone Dehydrogenase1 (NQO1) and glutathione S-transferase Pi 1 (GSTP1) [147]. Rutin favorably augmented antioxidant performance by mitigating ROS generation and malondialdehyde concentration in HepG2 cells [148]. In contrast, prolonged treatment of rutin caused a depletion of GSH in HepG2 cells and acted as a pro-oxidant, resulting in cell death [149]. Interestingly, rutin caused a significant cytotoxic effect on HepG2 cancer cells [150]. Rutin protected HepG2 cells against hydrogen peroxide induced DNA damage; however, it did not enhance the DNA repair process [122,123]. In an in vivo study, rutin hampered liver tumor markers, including α-fetoprotein and carcinoembryonic antigens, in nitrosodiethylamine and phenobarbital administered rats. Additionally, rutin enhances the declined level of membrane bound ATPases [151]. Na+/K+, Ca^2+^, and Mg^2+^ ATPases play a key role in the transportation of the electrolytes sodium, potassium, calcium, and magnesium across membranes [203]. The lipid peroxidation activity, which is often raised when in a cancerous state, plays a deleterious effect on ATPase activities and electrolyte levels [204], while electrolyte imbalance contributes to cancer progression [205]. Rutin reverses common electrolyte abnormalities, including hyperkalemia, hyponatremia, hypercalcemia, and hypomagnesemia in hepatocellular carcinoma-bearing rats [151]. Upregulation of enzymes involved in repairing DNA damage, including PARP, DNA polymerase β, and DNA ligase participate in tumorigenesis [206,207,208]. Modulation of these parameters is a promising way of controlling cancer. In an in vivo experiment, rutin interestingly hampered DNA damage and the activity of repair enzymes induced by hepatocarcinogens, namely aflatoxin B1 and N-nitrosodimethylamine [152].

### 5.7. Rutin and Gastric Cancer

Gastric cancer is a result of various genetic and environmental factors. *Helicobacter pylori* infection, smoking, dietary habits, and obesity are important risk factors influencing the development of gastric cancer [209]. According to the World Health Organization, gastric cancer is classified into three categories, such as adenocarcinoma, signet ring-cell carcinoma, and undifferentiated carcinoma [210]. Another most common classification system, the Laurén classification, categorized gastric cancer into two groups, namely intestinal and diffuse types [211]. Despite a multitude of advances achieved in treatment of gastric cancer, adverse effects and resistance to chemotherapeutic agents limit their therapeutic efficacy [212]. Therefore, new alternative strategies to overcome these challenges and the design of novel drugs for targeting gastric cancer therapy are needed. P38MAPK is a key factor in modulating various functions of tumor cells, including differentiation, invasion, proliferation, and apoptosis [213], attracting further interest as an auspicious therapeutic target for cancer therapy [214]. Rutin mitigates the proliferation of human gastric adenocarcinoma SGC-7901 cells, arrests tumor cells at G0/G1, upregulates caspase-3, caspase-7, and caspase-9, and lowers the Bcl-2/Bax ratio. These effects are attributed to the upregulation of the p38 signaling pathway. Concomitant treatment with rutin and oxaliplatin displays synergistic anticancer effects, allowing a decrease in the dose of oxaliplatin, thus decreasing toxicity [153].

### 5.8. Rutin and Prostate Cancer

Prostate cancer represents the second most commonly diagnosed cancer among men [215]. Prostate cancer is a result of both genetic and environmental factors; however, the main etiology is still unclear [216]. The combined use of chemotherapeutic drugs and nutraceutical agents is a promising solution for enhancing anticancer effects, as well as ameliorating drug resistance and chemotherapy adverse effects [217]. A combination of 5-fluorouracil (5-FU) and rutin synergistically acts as a potential cytotoxic agent against PC3 prostate cancer cells. Furthermore, combined treatment hampers cell proliferation, augments apoptosis, downregulates Bcl-2 signaling protein, and upregulates p53 expression [28]. Overactivation of Bcl-2 proto-oncogene plays a critical role in abrogating cell apoptosis and tumor suppressor protein p53 activity [61]. Further investigations should be conducted to evaluate the combined effects of rutin and 5-FU in the regulation of other pro-apoptotic signaling pathways. Voltage gated K^+^ channels (*I*K) participates in modulating numerous cellular activities related to cancer progression [218]. *I*K current inhibitors may be considered as suitable target of cancer therapy; however, George et al. [154] demonstrated that rutin presented no modulatory effect on *I*K current in human prostate cancer cell line (LNCaP). Future studies should be performed to evaluate the influence of novel anticancer compounds on *I*K current.

### 5.9. Rutin and Other Cancers

It has been well-stablished that augmentation of wingless/integrated (Wnt)/GSK-3β/β-catenin signaling pathway plays a key role in upregulation of P-gp in various cancer types [219]. Rutin enhanced doxorubicin-mediated cell cycle arrest at G2/M phase through interfering with Wnt/GSK-3β/β-catenin signaling pathway, thereby alleviated the overexpression of P-gp in drug resistant oral carcinoma KB cells [105]. Rutin-Cu (II) complex suppressed the growth and proliferation of cervical cancer cells (HeLa) in a time- and concentration-dependent manner [155]. Ovarian cancer is considered a second leading cause of gynecologic cancer death among women [220]. Although the chemotherapy and surgical procedures are applied in ovarian cancer therapy, the five-year survival rate is poor, less than 50% [221]. Rutin exerted an acceptable potential in abrogation of cell proliferation and VEGF expression of ovarian cancer OVCAR-3 cells [156]. Rutin also demonstrated antiangiogenic effects against B16F-10 melanoma cell-induced capillary formation in an animal model. In addition, rutin downregulated the expression of VEGF, IL-1β and enhanced the expression of TNF-α in tumor associated macrophage. Therefore, antiangiogenic activity of rutin can be attributed to the modulation of these cytokines and growth factors [157].

## 6. Nanostructured Formulations of Rutin in Combating Cancer

Despite the encouraging anticancer properties of rutin in preclinical studies, there are certainly obstacles in its clinical transition. Rutin has poor solubility, high metabolism, low gastrointestinal absorption, and limited bioavailability, therefore limiting the capability to achieve effective concentrations in tumor tissues [32,222]. A promising way to overcome these challenges is encapsulating the agent into various forms of nanosized delivery vehicles. Nanotechnology offers the potential to deliver bioactive phytochemicals and nutraceutical agents directly to the desired locations, such as tumor tissues, thereby providing maximum therapeutic activities of these compounds [19,223,224,225,226]. Nanostructured carriers can passively accumulate in solid tumors by an enhanced permeability retention effect [227,228]. Active targeting is attained by attachment of a targeting ligand to the nanoparticles (NPs) surface that binds to its receptor expressed on tumor cells, thereby increasing site-specificity and controlled drug delivery to the cancer tissue [229].

Rutin encapsulated in folic acid conjugated keratin NPs promotes cell death in MCF-7 breast cancer cells while exhibiting less toxicity in healthy cells [230] (Table 2). Additionally, the actively targeted nanoformulation decreases tumor cell migration, elevates rutin uptake in cancer cells, and boosts apoptosis through ROS production and mitochondrial potential loss. An in vitro study indicated that the nanoformulation selectively targets breast cancer cells [230]. However, in vivo studies are needed to confirm the active targeted delivery of folic acid conjugated keratin NPs.

Nanoemulsions are thermodynamically stable systems that are favorable vehicles for enhancing solubility, intestinal uptake, and bioavailability of lipophilic drugs [231,232]. Rutin-based nanoemulsion dramatically promotes cytotoxicity in PC3 prostatic cancer cells through inducing ROS and apoptosis. The rutin nanoemulsion is more effective against prostate cancer when compared to rutin suspension. Optimized rutin nanoemulsion exhibits thermodynamic stability and an efficient drug release profile [233]. Rutin nanoemulsion may be a suitable candidate to be evaluated in in vivo models of prostate cancer.

Ionic liquids are salts with a melting point below 100 °C which enhance the solubility of poorly water-soluble drugs [245,246]. Ionic liquids are composed of organic cations, such as imidazolium, pyrrolidinium, pyridinium, tetraalkylammonium, or tetraalkylphosphonium, along with organic or inorganic anions, including tetrafluoroborate, hexafluorophosphate, and bromide [247]. Interestingly, hybrid ionic liquids contain active pharmaceutical ingredients and ionic liquids, which are promising strategies to improve their solubility, bioavailability, and biological effects. For instance, ionic liquid-based formulations enhanced the solubility and anticancer activities of several compounds, such as curcumin and paclitaxel [248,249]. Rutin-loaded ionic liquid–NPs were fabricated by a double-emulsion method and were found to exhibit cytotoxic effects against 786-O human renal cancer cells through amplifying sub-G1 population. Ionic liquids increased the solubility of rutin and enhanced its incorporation into water/oil/water emulsion, thereby providing a controlled delivery system [234].

The development of stimuli-responsive nanocarriers is another promising solution for the targeted delivery and site-specific triggering of the release of anticancer agents [250,251]. Stimuli-sensitive nanocarriers rapidly release anticancer drugs in response to environmental stimuli, such as pH, temperature, redux, and enzymes [252,253]. Eudragit S100 is a pH-sensitive copolymer that dissolves at colon pH and is extensively engaged for drug targeting to the colon [254,255]. Rutin-loaded eudragit S100 nanospheres display pH-sensitive activity that can effectively achieve rutin into the colon. The pH-sensitive nanospheres significantly increase the solubility of rutin and provoke its cytotoxic activity against human colon cancer HCT 116 cells vs. rutin suspension [235]. Biodistribution and in vivo studies should be conducted to better understand the anticancer potential of rutin-loaded pH-sensitive nanospheres.

As another delivery system, protein-based NPs possess certain advantages, since they are inherently biocompatible, stable, and have a potential for surface functionalization and covalent attachment of ligands for targeted drug delivery [256]. Keratin is a natural protein that is abundantly found in human hair [257]. Biocompatible and stable keratin NPs incorporated with dual phytocompounds, rutin and quercetin, were successfully fabricated. According to an in silico study, the keratin-based NPs eagerly dock into binding pockets of H-Ras P21 proto-oncogene. This report has been supported by an in vitro study in which the nanoformulation caused significant cytotoxicity in Hela cervical cancer cells [236].

Consistently, poly (lactic-co-glycolic acid) (PLGA) NPs also attain special attention in biomedical applications as they represent desirable features, including biocompatibility, surface modifiability, controlled delivery, and targeting [258]. Oral administration of rutin-loaded PLGA NPs ameliorates diethylnitrosamine-induced HCC. This beneficial effect is mediated by decreasing pro-inflammatory cytokines, including IL-1β, TNF-α, and IL-6, as well as abrogating the NF-κB inflammatory cascade. From another mechanistic point of view, the nanoformulation restores membrane-bound enzymes and mitigates the enhanced level of hepatic enzymatic and α-glutamyl transferase (GGT). PLGA NPs also enhance endogenous antioxidant activity (confirmed by increasing the content of SOD, CAT, GSH, and GPx), suggesting its protective effect against HCC [237].

In addition to the previous goals, nanocarriers offer a potential strategy for efficient delivery of combination anticancer drugs to overcome MDR and decrease the frequency of drug administration during combination therapy of anticancer agents [259,260]. The co-delivery of benzamide along with rutin through PLGA nanospheres synergistically suppresses the proliferation of MDA-MB-231 cells in the G0/G1 phase through empowering apoptosis and ROS generation. The polymeric nanospheres provide a sustained release of chemotherapeutic agents, augment therapeutic efficiency, and target a MDR associated phenotype TNBC [238]. Chang et al. [239] revealed that rutin–chitosan nanoconjugates could promisingly induce apoptosis and cell cycle arrest in TNBC. The fabricated nanoconjugates markedly stop TNBC growth at a concentration of 12.5 µg/mL.

In recent years, metal-based biocompatible NPs have attracted scientific interest as they are cost-effective, eco-friendly, easy to synthesize, and simple to modify and functionalize the surface [261,262]. Metal NPs possess various applications in the treatment of several diseases, including cancer [263]. Biosynthesized zinc oxide (ZnO) NPs using rutin display higher cytotoxic effect against MCF-7 breast cancer cells vs. rutin alone [240]. In another study, chitosan functionalized copper oxide (CuO) nanocomposites were biosynthesized using rutin and exhibited antiproliferative activity, provoking apoptosis in the human lung cancer cell line A549 [241]. Further mechanistic studies are needed to confirm the promising effects of rutin-loaded metal NPs in the treatment of cancer.

Fucoidan, a natural sulfated polysaccharide, possesses widespread applications in the treatment of cancer, inflammatory disease, and bacterial infections [264,265,266]. Fucoidan is able to form complexes with different drugs using reactive functional groups to increase their solubility, absorption, and bioavailability [267,268]. The nanosized rutin-fucoidan complex is biocompatible in normal cells and provides sustained release of compounds from the mixture at a pH of 5.5. Additionally, the complex synergistically boosts growth effects, arrests the cell cycle, and enhances apoptosis through ROS production, mitochondrial potential loss, and DNA fragmentation in HeLa cervical cancer cells [242]. The development of such complex formulations can be considered as a promising solution to counteract cervical cancer, amongst other cancer types. Rutin-loaded chitosan/poly (acrylic acid) nanogel enhanced bioavailability of rutin and significantly reinforced antiproliferative, antiangiogenic (by reducing VEGF), and apoptotic effects (by increasing p53, caspase-3, and Bax as well as mitigating Bcl-2), indicating potential antitumor activity of the nanoformolation against diethylnitrosamine (DENA)/carbon tetrachloride (CCl_4_)-induced hepatocarcinoma in rats [243]. However, both free rutin and rutin-loaded nanosized polymeric micelles displayed low cytotoxicity in sensitive K562 and resistant K562/ADR cells [244].

Overall, experimental results demonstrate that the properties of nano drug delivery systems have been able to overcome pharmacokinetic limitations of rutin, underscoring its promising effects in chemotherapy. Further research needs to be performed to design surface-modified nanoformulations of rutin to attain optimized drug delivery systems. Various novel drug delivery systems of rutin and their effects on improving pharmacokinetic limitations are depicted in Figure 3.

## 7. Conclusions

The plant kingdom offers a tremendous source of alternative anticancer drugs. Among natural entities, rutin, a glycosylated flavonoid, possesses several significant biological activities with the prevailing evidence now being focused on its anticancer effects. Rutin has been shown to employ multiple mechanisms to impede cancer initiation and progression by modulating various dysregulated signaling pathways implicated in inflammation, apoptosis, autophagy, and angiogenesis (Figure 4). Specifically, the tumor-inhibitory effects of rutin have been shown to be exerted through the regulation of various signaling pathways, such as PI3K/Akt/mTOR, NF-κB, Nrf2, ERK, p38 MAPK, and JNK. This bioactive natural agent potentially interferes with several intracellular signaling molecules, including TNF-α, ILs, LC3/Beclin, Bax, Bcl-2, caspases, and VEGF. In particular, extensive studies have revealed that rutin targets various therapeutically important molecules, such as p53, Bax, Bcl-2, caspase-3, caspase-9, NF-κB, Akt, TNF-α, Atg5, Beclin, GSH, and SOD (Figure 5). Several cancer types, including breast cancer, glioblastoma, prostate cancer, lung adenocarcinoma, gastric cancer, hepatocellular carcinoma, leukemia, and colon cancer, are impacted by rutin. Most of the current anticancer evidence of rutin is focused on in vitro models of cancer, with very limited in vivo studies. Despite various preclinical mechanistic studies on the anticancer effects of rutin, lack of well-designed randomized clinical trials on the therapeutic activities and safety of rutin escalates the need toward more clinical investigations. The possible pharmacokinetic limitations of rutin underscore the need for developing appropriate delivery systems. Additional studies and engineering methods are required to design surface modified nanostructures of rutin to achieve targeted drug delivery systems against cancer. A further area of research on novel molecular targets and signaling pathways of rutin, as well as providing well-controlled clinical trials, will develop its clinical applications in the prevention and treatment of several cancer types.

## Figures and Tables

**Figure 1 cancers-12-02276-f001:**
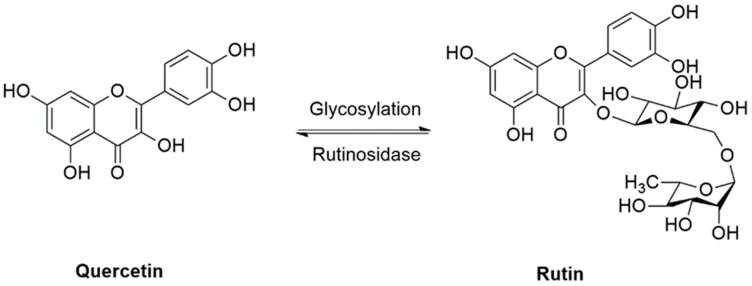
Rutin, a glycoside from quercetin flavonoid.

**Figure 2 cancers-12-02276-f002:**
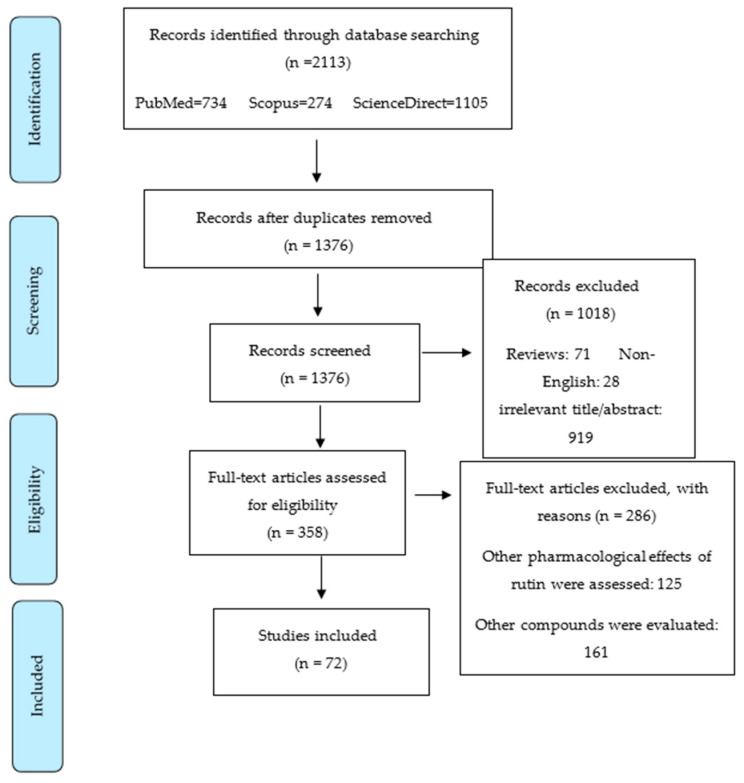
Flow diagram related to selection process of articles.

**Figure 3 cancers-12-02276-f003:**
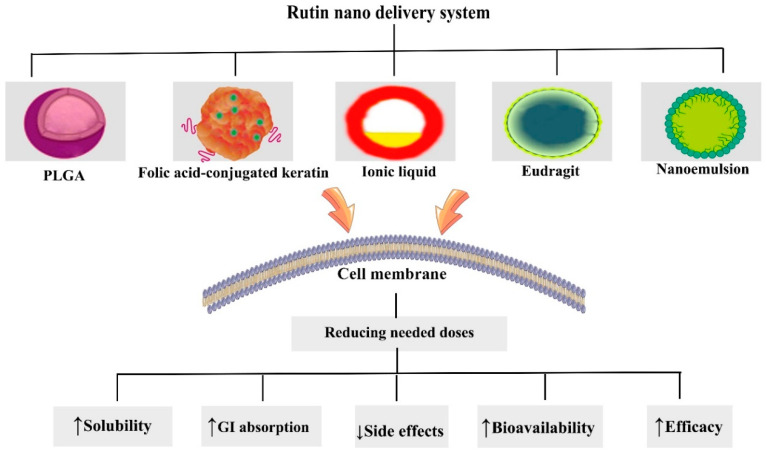
Nanoformulations of rutin used to combat cancer. GI, gastrointestinal; PLGA, poly (lactic co-glycolic acid).

**Figure 4 cancers-12-02276-f004:**
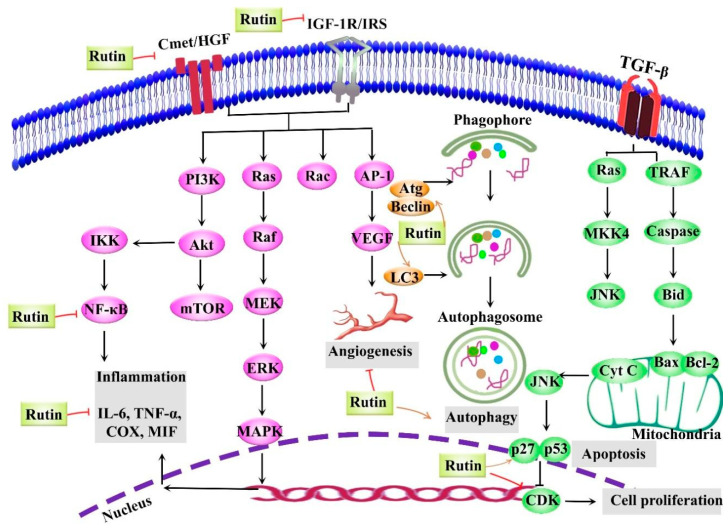
Numerous dysregulated mechanisms and therapeutic targets implicated in the anticancer effects of rutin. Akt, protein kinase B; Atg, autophagy-related gene; Bax, Bcl-2-associated X protein; Bcl-2, B-cell lymphoma 2; CDK, cyclin-dependent kinase; C-met/HGF, mesenchymal–epithelial transition factor/hepatocyte growth factor; COX, cyclooxygenase; Cyt C, cytochrome c; ERK, extracellular signal-regulated kinase; IGF-1R/IRS-1, insulin-like growth factor 1 receptor /insulin receptor substrate-1; IKK, IκB kinase; IL-6, interleukin-6; JNK, Jun N-terminal Kinase; LC3, light chain 3; MAPK, mitogen-activated protein kinase; MEK, mitogen-activated protein kinase kinase; MIF, macrophage migration inhibitory factor; MKK, MAPK kinase; mTOR, mammalian target of rapamycin; NF-κB, nuclear factor-κB; PI3K, phosphatidylinositol 3-Kinase; TGF-β, transforming growth factor-β; TNF-α, tumor necrosis factor-α; TRAF, tumor necrosis factor receptor-associated factor; VEGF, vascular endothelial growth factor.

**Figure 5 cancers-12-02276-f005:**
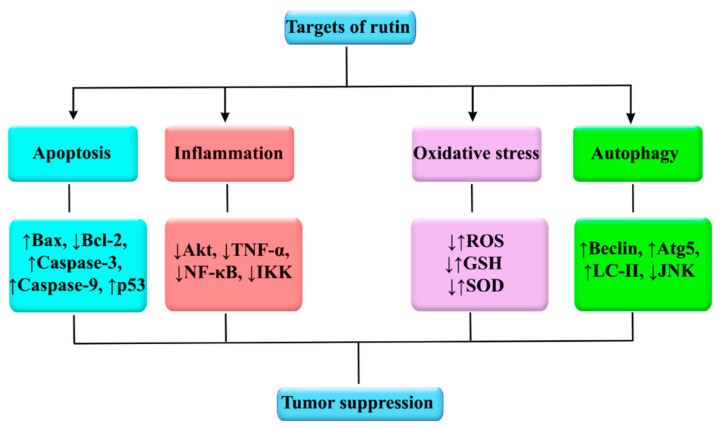
Main molecular targets influenced by rutin in cancer. Akt, protein kinase B; Atg, autophagy-related gene; Bax, Bcl-2-associated X protein; Bcl-2, B-cell lymphoma 2; GSH, glutathione; IKK, IκB kinase; JNK, Jun N-terminal Kinase; LC, light chain; NF-κB, nuclear factor-κB; ROS, reactive oxygen species; SOD, superoxide dismutase; TNF-α, tumor necrosis factor-α.

**Table 1 cancers-12-02276-t001:** Potential anticancer effects and mechanisms of action of rutin based on in vitro and in vivo studies.

Type of Cancer	Type of Study	Cell Type/Animal Model	Anticancer Effects	References
Breast	In vitro In vivo	Human TNBC cells (MDA-MB-231 and MDA-MB-468) Female athymic Foxn1nu/Foxn1C mice	↓c-met/ HGF, ↓paxillin, ↓Rac-1 ↓mTOR, ↓Akt, ↓tumor volume	[101]
Breast	In vitro	Human breast cancer cells (MCF-7)	↓Proliferation, ↑apoptosis, ↑cell cycle arrest, ↑PTEN, ↑p53 ↑p21	[102]
Breast	In vitro In vivo	Human breast cancer cells (MCF-7) Female Swiss albino mice	↑Apoptosis, ↓tumor volume, ↓CEA, ↓cholesterol, ↓FAS, ↓MDA, ↑GSH, ↑caspase-3, ↑caspase-7	[103]
Breast	In vitro	Human TNBC cells (MDA-MB-231) and breast cancer cells (MCF-7)	↑Chemosensitivity, ↓MDR, ↓P-gp ↓BCRP	[104]
Breast	In vitro	Human breast cancer cells (MCF-7)	↑Chemosensitivity	[105]
Breast	In vitro	Human TNBC cells (MDA-MB-231)	↓VEGF, ↓angiogenesis	[106]
Breast	In vitro	Human breast cancer cells (MCF-7)	↑Apoptosis, ↑cell cycle arrest	[107]
Breast	In vitro	Human TNBC cells (MDA-MB-231) and Human breast cancer cells (MCF-7)	↑Apoptosis, ↑p53, ↑Bax, ↓Bcl-2, ↓VEGF	[108]
Breast	In vitro	Human TNBC cells (MDA-MB-231) and human breast cancer cells (MCF-7)	↓Proliferation, ↑apoptosis, ↑ROS	[109]
Lung	In vitro	Human lung cancer cells (A549)	↑Cytotoxicity, ↑GSK-3β, ↑TNF-α	[110]
Lung	In vitro	Human lung cancer cells (A549)	↓Migration, ↓fibronectin, ↓collagen type I and IV, ↑ROS, ↓superoxide	[29]
Lung	In vitro	Human lung cancer cells (A549)	↓Single strand DNA break, ↓ROS	[111]
Lung	In vivo	C57BL/6 female mice	↓Lung tumor nodules, ↑life span	[112]
Lung	In vivo	Albino Swiss mice	↓Lung tumor nodules, ↓growth, ↓invasion index	[113]
Lung	In vitro	Human lung cancer cells (A549)	↑Autophagy, ↑Beclin1, ↑Atg5/12, ↑LC3-II, ↓NF-κB, ↓TNF-α	[114]
Lung	In vitro	Human lung cancer cells	↓Proliferation ↓cell cycle, ↓NF-κB, ↓p38	[115]
Colon	In vitro	Human colon cancer cells (HT-29)	↑Apoptosis, ↑caspase-3, ↑caspase-8, ↑caspase-9 ↑PARP, ↓Bcl-2, ↑Bax	[116]
Colon	In vitro In vivo	Human colon cancer cells (SW480) nu/nu mice	↓Tumor growth ↓angiogenesis, ↓VEGF	[117]
Colon	In vitro	Human colon cancer cells (HT-29)	↑Cytotoxicity, ↓mitochondrial membrane potential, ↑lipid peroxidation, ↓SOD ↓CAT ↓GPx	[118]
Colon	In vitro	Human colon cancer cells (HT-29)	↓Adhesion, ↓migration, ↑ROS, ↓superoxide	[29]
Colon	In vitro	Human colon cancer cells (HT-29)	↑Apoptosis, ↓Bcl-2, ↑Bax, ↑caspase-3, ↑caspases-8, ↑caspase-9, ↑p53, ↓NF-kB, ↓IKK-α, ↓IKK-β, ↓MAPK	[119]
Colon	In vitro	Human colon cancer cells (SW480)	↑Apoptosis, ↑cell cycle arrest, ↓metabolism	[120]
Colon	In vitro	Human colon cancer cells (HT-29)	↓ cell viability	[121]
Colon	In vitro	Human colon cancer cells (LoVo)	↑Apoptosis, ↑cell cycle arrest	[107]
Colon	In vitro	Human colon cancer cells (Caco2)	↓DNA damage	[122]
Colon	In vitro	Human colon cancer cells (Caco2)	No effect on DNA repair	[123]
Colon	In vivo	Female CF1 mice	↓Focal areas of dysplasia, ↓hyperproliferation	[124]
Colon	In vivo	Male F344 rats	↓Aberrant crypt foci, ↑apoptosis	[125]
Colon	In vivo	Male F344 rats	No effect	[126]
Colon	In vitro	Human colon cancer cells (HCT-8)	No effect	[127]
Colon	In vivo	Male F344 rats	No effect	[128]
Brain	In vitro	Human glioblastoma cell line (GL-15)	↓Proliferation, ↑apoptosis, ↓ERK ↑GFAP	[129]
Brain	In vitro	Human glioblastoma cell line (GL-15)	↓Invasion, ↓angiogenesis, ↓VEGF, ↓TGF-β1	[130]
Brain	In vitro	Human glioblastoma cell line (GL-15)	↓Proliferation, ↓invasion, ↓MMP-2, ↑fibronectin, ↑laminin	[131]
Brain	In vitro	Human neuroblastoma cells (LAN-5)	↑Apoptosis, ↓cell cycle, ↓TNF-α, ↓Bcl-2, ↑Bax	[132]
Brain	In vitro	Human glioma cells (CHME)	↑p53, ↑caspase-3, ↑caspase-9, ↑cytochrome c, ↑Bax, ↓Bcl-2, ↑ROS ↓mitochondrial membrane potential	[133]
Brain	In vitro In vivo	Human glioblastoma cells (U87-MG, D54-MG, and U251-MG) BALB/c athymic mice	↑Cytotoxicity, ↑apoptosis, ↓ JNK, ↓autophagy, ↑caspase-3,	[134]
Leukemia	In vitro	Human leukemic cells (U937, HL-60, KG1, and KG1a)	↑Cytotoxicity, ↑apoptosis, ↓GSK-3β, ↑Akt	[135]
Leukemia	In vivo	Human leukemia HL-60 cells induced leukemia in BALB/c mice	↓Tumor weight, ↓tumor volume	[136]
Leukemia	In vivo	Murine leukemia WEHI-3 cells induced leukemia in BALB/c mice	↓Proliferation, ↓macrophage phagocytosis	[137]
Leukemia	In vitro	Human leukemic cells (THP-1)	↑Autophagy, ↓NF-κB, ↓TNF-α	[114]
Leukemia	In vitro	Human promyelocytic leukemia cells (HL-60)	↓Angiogenesis, ↓VEGF, ↓AP-1, ↓IGF-1R/IRS-1	[138]
Leukemia	In vitro	Human acute myeloid leukemia cells (KG1)	↑Cytotoxicity, ↑antioxidant activity	[139]
Leukemia	In vitro	human myelogenous leukemia cells (K562)	↓Single strand DNA break, ↓ROS	[140]
Leukemia	In vitro	human myelogenous leukemia cells (K562)	↑Apoptosis	[141]
Leukemia	In vitro	human promyeloleukemic cells (HL-60)	No effect	[142]
Leukemia	In vitro	Murine leukemia cells (L1210)	No effect	[143]
Multiple myeloma	In vitro	Human multiple myeloma cells (RPMI8226)	↑Cytotoxicity, ↑antioxidant activity	[139]
Multiple myeloma	In vitro	Human multiple myeloma cells (ARH–77)	↑Cytotoxicity, ↓mitochondrial and lysosomal activity	[144]
Lymphoma	In vitro	Dalton’s lymphoma cells	↑Apoptosis, ↓Bcl-xL, ↓c-FLIP, ↓GST, ↓GR	[145]
Liver	In vitro	Rat hepatoma cells (HTC)	↓Proliferation, ↓cell viability	[146]
Liver	In vitro	Human liver cancer cells (HEPG2)	↓Proliferation, ↑apoptosis, ↓CYP3A4, ↓CYP1A1, ↑NQO1, ↑GSTP1	[147]
Liver	In vitro	human hepatoma cell line (HepG2)	↓ROS, ↓MDA	[148]
Liver	In vitro	human hepatoma cell line (HepG2)	↓GSH	[149]
Liver	In vitro	human hepatoma cell line (HepG2)	↑Cytotoxicity	[150]
Liver	In vitro	human hepatoma cell line (HepG2)	↓DNA damage	[122]
Liver	In vitro	human hepatoma cell line (HepG2)	No effect	[123]
Liver	In vivo	Wistar albino rats	↑Membrane bound ATPases	[151]
Liver	In vivo	Wistar rats	↓PARP, ↓DNA polymerase β, ↓DNA ligase	[152]
Gastric	In vitro	Human gastric cancer cells (SGC-7901)	↑Apoptosis, ↑caspase-3, ↑caspase-7, ↑caspase-9, ↓Bcl-2/Bax, ↑p38MAPK, ↑G0/G1 arrest	[153]
Prostate	In vitro	Human prostatic cancer cells (PC3)	↓Proliferation, ↑apoptosis, ↓Bcl-2, ↑p53	[28]
Prostate	In vitro	Human prostate cancer cells (LNCaP)	No effect	[154]
Oral	In vitro	Drug resistance oral carcinoma cells (KBCH^R^8–5)	↓Wnt/GSK-3β/β-catenin pathway, ↓P-gp	[105]
Cervical	In vitro	cervical cancer cells (HeLa)	↓Proliferation, ↓growth	[155]
Ovarian	In vitro	ovarian cancer cells (OVCAR-3)	↓Proliferation, ↓VEGF	[156]
Melanoma	In vitro	melanoma cells (B16F-10)	↓Angiogenesis, ↓VEGF, ↓IL-1β, ↑TNF-α	[157]

Abbreviations: Akt, protein kinase B; AP-1, activating protein-1; Atg5/12, autophagy related 5/12; Bax, Bcl-2 associated X protein; Bcl-2, B cell lymphoma 2; BCRP, breast cancer resistance protein; CAT, catalase; CEA, carcinoembryonic antigen; c-FLIP, cellular FLICE-inhibitory protein; C-met, mesenchymal–epithelial transition factor; CYPs, cytochrome P450s; FAS, fatty acid synthase; GFAP, glial fibrillary acidic protein; GPx, glutathione peroxidase; GR, glutathione reductase; GSK-3β, glycogen synthase kinase; GST, glutathione S-transferase; GSTP1, glutathione S-transferase Pi 1; HGF, hepatocyte growth factor; IGF-1R, insulin-like growth factor-1 receptor; IKK, IκB kinase; IL, interleukin; IRS-1; insulin receptor substrate-1; JNK, Jun N-terminal Kinase; LC3-II, light chain 3; MAPK, mitogen-activated protein kinase; MDR, multidrug resistance; MMP-2, metalloproteinase; mTOR, mammalian target of rapamycin; NF-κB, nuclear factor-κB; NQO1, NADPH quinone oxidoreductase 1; PARP, poly (ADP ribose) polymerase; P-gp, P-glycoprotein; PTEN, Phosphatases and tensin homolog; Rac-1, Ras-related C3 botulinum toxin substrate 1; ROS, reactive oxygen species; SOD, superoxide dismutase; TGF-β, transforming growth factor-β; TNBC, triple-negative breast cancer; TNF-α, tumor necrosis factor-α; VEGF, vascular endothelial growth factor; Wnt, wingless/integrated.

**Table 2 cancers-12-02276-t002:** Rutin based nanoscale drug delivery systems for counteracting several types of cancer.

Nanoformulation Model	Type of Cancer	Type of Study	Cell Type/Animal Model	Outcomes	References
Folic acid-conjugated keratin NPs	Breast	In vitro	Huma breast cancer cells (MCF-7)	↑Apoptosis, ↓migration, ↑ROS, ↓mitochondrial membrane potential	[230]
Nanoemulsions	Prostate	In vitro	Human prostatic cancer cells (PC3)	↑Apoptosis, ↑ROS	[233]
Ionic liquids-NPs	Renal	In vitro	Human renal cancer cells (786-O)	↑Cytotoxicity, ↑sub-G1 population, ↑solubility	[234]
Eudragit S100 nanospheres	Colon	In vitro	Human colon cancer cells (HCT 116)	↑Cytotoxicity ↑solubility	[235]
Keratin NPs	Cervical	In vitro	Human cervical cancer cells (Hela)	↑Cytotoxicity	[236]
PLGA NPs	Liver	In vivo	Albino male Wistar rats	↓IL-1β, ↓TNF-α, ↓IL-6 ↓NF-κB, ↑SOD, ↑CAT, ↑GSH, ↑GPx, ↑membrane-bound enzymes	[237]
PLGA nanospheres	Breast	In vitro	Human TNBC cells (MDA-MB-231)	↓Proliferation, ↑apoptosis, ↑ROS	[238]
Chitosan NPs	Breast	In vitro	Human TNBC cells (MDA-MB-231)	↑Apoptosis, ↑cell cycle arrest	[239]
ZnO NPs	Breast	In vitro	Human breast cancer cells (MCF-7)	↑Cytotoxicity	[240]
Chitosan/copper oxide nanocomposites	Lung	In vitro	Human lung cancer cells (A549)	↑Cytotoxicity, ↑apoptosis	[241]
Fucoidan NPs	Cervical	In vitro	Human cervical cancer cells (Hela)	↑DNA fragmentation, ↑cell cycle arrest, ↑ROS, ↓mitochondrial membrane potential	[242]
Chitosan/poly (acrylic acid) nanogel	Liver	In vivo	Male albino rats	↓proliferation, ↓angiogenesis, ↓VEGF, ↑Bax, ↓Bcl-2, ↑p53, ↑caspase-3	[243]
Nanosized polymeric micelles	Leukemia	In vitro	human myelogenous leukemia cells (K562)	Low cytotoxicity	[244]

Abbreviations: Bax, Bcl-2 associated X protein; Bcl-2, B cell lymphoma 2; CAT, catalase; GPx, glutathione peroxidase; GSH, glutathione; IL-6, interleukin-6; NF-κB, nuclear factor-κB; NPs, nanoparticles; PLGA, poly (lactic co-glycolic acid); ROS, reactive oxygen species; SOD, superoxide dismutase; TNF-α, tumor necrosis factor-α; VEGF, vascular endothelial growth factor.

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
