# Peer review of "Targeting Multiple Signaling Pathways in Cancer: The Rutin Therapeutic Approach"

_cancers, 2020, doi:10.3390/cancers12082276_

Round 1

Reviewer 1 Report

Dear Authors,

The review manuscript "Targeting Multiple Signaling Pathways in Cancer: The Rutin Therapeutic Approach" is interesting, is written in a correct English and reports novel and promising findings based on properly selected, 72 experimental reports. The article structure and the discussion are appropriate. Moreover, the article describes in a detailed and well-structured manner the aspects of rutin influence on the molecular processes involved in the development of various types of cancer, and highlights its chemopreventive properties. In addition, it also takes into account the importance of nanostructured formulations of rutin in ADME processes. The tables and figures are also carefully and clearly prepared. Summing up, I recommend to publish this article after checking for possible grammar/spelling typos.

Kind regards

Author Response

Comment 1:

The review manuscript "Targeting Multiple Signaling Pathways in Cancer: The Rutin Therapeutic Approach" is interesting, is written in a correct English and reports novel and promising findings based on properly selected, 72 experimental reports. The article structure and the discussion are appropriate. Moreover, the article describes in a detailed and well-structured manner the aspects of rutin influence on the molecular processes involved in the development of various types of cancer, and highlights its chemopreventive properties. In addition, it also takes into account the importance of nanostructured formulations of rutin in ADME processes. The tables and figures are also carefully and clearly prepared. Summing up, I recommend to publish this article after checking for possible grammar/spelling typos.

Response:

We greatly appreciate the respected reviewer for the generous comments regarding the quality of our work. According to the valuable comments, we have carefully rechecked the manuscript for any grammatical and spelling errors.

Reviewer 2 Report

Please find attached document

Author Response

General comments:

This review present therapeutic potential of rutin in cancer treatment and in targeting Multiple Signaling Pathways in Cancer. In my opinion review is well written, however minor revision is required before publication.

Response:

We express our sincere thanks to the respected reviewer for the encouraging comment about the quality of our manuscript. We have revised our manuscript as described below.

Comment 1:

Section about gastric cancer should be extended by classification of gastric cancer and rutin assisted treatment approach (if there is one)

Response:

According to your valuable comment, the classification of gastric cancer has been added to “Rutin and Gastric Cancer” section with appropriate references (page 15, lines 551-554).

Comment 2:

What about rutin supplentation when a recurrent tumor is identified?

Response:

With thanks for your valuable comment, we have added descriptions of the promising role of rutin in recurrent tumor management and cited appropriate references (page 10, line 321 to page 11, line 328).

Comment 3:

Could the authors explain in detail why citostatics from natural sources are less toxic that synthetised ones?

Response:

We have added a paragraph to the “Introduction” section (page 2, lines 70-79) to provide the explanation. 

Comment 4:

I have some concern about short introduction dealing with ionic liquids. The authors wrote that ILs are organic salt that can be used for cancer treatment. However, usually one component of IL is organic, and second one is inorganic. Moreover, I suggest to present specific examples of IL application in cancer therapy than write overall comments about ILs.

Response:

Thanks for your valuable suggestions. The short introduction dealing with ionic liquids has been modified. Additionally, we have provided specific examples of the application of ionic liquids in cancer therapy (page 17, line 630 to page 18 line 641). 

Reviewer 3 Report

This article thoroughly summerizes the large body of scientific literature on the compound rutin and its effect on cancer growth.

The challenge is that there are no clinical trials to date and thus lack of data on toxicities what might be important when using such an non-defined substance. Thus, the molecular mechanism behind therapy response of this compound is not understood. Although there are multiple signaling pathways described that are affected by this drug, direct ligands are not clearly characterized. This is one aspect that could be addressed in greater detail - eg in a table.

Due to the large amount of publications on this drug, this review is a nice addition to get an overview about this field of research.

Author Response

Comment 1:

This article thoroughly summarizes the large body of scientific literature on the compound rutin and its effect on cancer growth.

The challenge is that there are no clinical trials to date and thus lack of data on toxicities what might be important when using such a non-defined substance. Thus, the molecular mechanism behind therapy response of this compound is not understood. Although there are multiple signaling pathways described that are affected by this drug, direct ligands are not clearly characterized. This is one aspect that could be addressed in greater detail - eg in a table.

Due to the large amount of publications on this drug, this review is a nice addition to get an overview about this field of research.

Response:

We greatly appreciate the reviewer’s generous comments and constructive suggestions. We have added a new figure (Figure 5) and accompanying text to discuss the main molecular targets of rutin (page 20, lines 719-721).